# Seeing the advantage: visually grounding word embeddings to better capture human semantic knowledge

**Danny Merkx**
Radboud University
Nijmegen, The Netherlands
danny.merkx@ru.nl

**Stefan L. Frank**
Radboud University
Nijmegen, The Netherlands
stefan.frank@ru.nl

**Mirjam Ernestus**
Radboud University
Nijmegen, The Netherlands
mirjam.ernestus@ru.nl

## Abstract

Distributional semantic models capture word-level meaning that is useful in many natural language processing tasks and have even been shown to capture cognitive aspects of word meaning. The majority of these models are purely text based, even though the human sensory experience is much richer. In this paper we create visually grounded word embeddings by combining English text and images and compare them to popular text-based methods, to see if visual information allows our model to better capture cognitive aspects of word meaning. Our analysis shows that visually grounded embedding similarities are more predictive of the human reaction times in a large priming experiment than the purely text-based embeddings. The visually grounded embeddings also correlate well with human word similarity ratings. Importantly, in both experiments we show that the grounded embeddings account for a unique portion of explained variance, even when we include text-based embeddings trained on huge corpora. This shows that visual grounding allows our model to capture information that cannot be extracted using text as the only source of information.

## 1 Introduction

Distributional semantic models create word representations that quantify word meaning based on the idea that a word's meaning depends on the contexts in which the word appears. Such representations (also called embeddings) are widely used as the linguistic input for computational linguistic models, with research showing that they can account for response times in lexical decision tasks (Mandera et al., 2017; Rotaru et al., 2018; Petilli et al., 2021), decode brain data (Xu et al., 2016; Abnar et al., 2018), account for brain activity during text comprehension (Frank and Willems, 2017), and correlate with human judgements of word similarity (Kiela et al., 2018; Derby et al., 2018, 2020).

While such embeddings have proven useful, they are not cognitively plausible as creating high quality embeddings requires billions of word tokens. For instance, the GloVe embeddings developed by Pennington et al. (2014) are trained on 840 billion words. It would require a human 80 years of constant reading at about 330 words per second to digest that much information. Obviously, humans are able to understand language after much less exposure, and furthermore, their sensory experience is much richer than solely reading texts.

Embodied cognition theory poses that our conceptual knowledge is based on the entirety of our sensory experience (Barsalou, 2008; Foglia and Wilson, 2013). For instance, reading the word *dog* elicits sensory experiences we have with dogs, such as their sound and how they look. Embodied cognition theory thus assumes that all our sensory experiences contribute to our conceptual knowledge and processing, which should be reflected in human behaviour. Early priming studies have indeed found that visual similarities can elicit priming effects (D'Arcais et al., 1985; Schreuder et al., 1998).

If visual features are part of our conceptual knowledge, word embeddings incorporating visual features should be able to explain human behavioural data to a degree unattainable by purely text-based methods (that is, if we assume visual sensory experiences can never be fully captured by textual descriptions). That is why recent research has taken an interest in multimodal word embeddings, combining text with a second source of information, resulting in visually grounded embeddings (VGEs) in the case of visual information.

### 1.1 Related work

Using image tags as a source of visual context, Bruni et al. (2013) create visual distributional semantic embeddings and use dimensionality reduction to map visual and text-based embeddings to the common VGE space. Derby et al. (2018) combine

text-based embeddings with the network activations of an object recognition model and show that these visual features improve the embeddings' performance in downstream tasks. Petilli et al. (2021) use visual embeddings created by an object recognition network, and show that the embedding similarities are predictive of priming effects over and above text-based similarities.

The studies described above involve separately trained word and visual embeddings. An end-to-end approach to combine visual and linguistic information is through a deep neural network based caption-to-image retrieval (C2I) models (e.g., Karpathy and Fei-Fei 2015; Kamper et al. 2017). While these models are trained to encode images and corresponding written or spoken captions in a common embedding space such that relevant captions can be retrieved given an image and vice versa, the resulting embeddings have been shown to capture sentence-level semantics (Chrupała et al., 2017; Merkx and Frank, 2019; Merkx et al., 2021). Kiela et al. (2018) showed that pretrained embeddings correlated better with human intuition about word meaning after being fine-tuned as learnable parameters in their C2I model.

## 1.2 Current study

In this study we investigate whether VGEs created by a C2I model explain human behavioural data. Our research question is: can VGEs capture aspects of word meaning that (current) text-based approaches cannot? To answer this question we investigate novel end-to-end trained VGEs and test them on two types of human behavioural data thought to rely on conceptual/semantic knowledge. Secondly, we take care to separate the contribution of the image modality from that of the linguistic information to see whether visual grounding captures word properties that cannot be learned by purely text-based methods. We do this by comparing our VGEs to three well-known text-based methods.

Throughout our experiments we will use two versions of the text-based methods: custom trained on the same data as our VGEs and pretrained on large corpora. From a cognitive modelling perspective, the former of these is more interesting. While the use of large corpora may not be problematic for natural language processing applications where performance comes first, we aim to create cognitively plausible embeddings, that is, from a realistic amount of linguistic exposure. However, the inclu-

sion of pretrained embeddings serves to answer our main research question.

### 1.2.1 Semantic similarity judgements

In our first experiment we test whether the VGEs correlate better with a measure of human intuition about word meaning than text-based embeddings. A well-known method to capture human intuition about word meaning is simply by asking subjects how similar two words are in meaning. To evaluate word embeddings, one can then see if embedding similarities for those word pairs correlate with the human judgements (e.g., Bruni et al., 2013; Baroni et al., 2014; Speer and Chin, 2016; Kiela et al., 2018; Derby et al., 2020).

While the study by Kiela et al. (2018) performed a similar investigation on pretrained word embeddings fine-tuned through their C2I model, they did not take into account the fact that text might also contain visual knowledge. It is not unreasonable to assume that some visual knowledge can be gained from a large corpus of sentences solely describing visual scenes. We account for this visual knowledge from text by incorporating word embeddings trained on the image descriptions in order to investigate the contribution of the *image* modality included in the VGEs.

Collecting word similarity ratings typically involves showing participants two words and asking them to rate how similar or related their meanings are, or picking the most related out of several pairs. Semantic relatedness refers to the strength of the association between two word meanings. For instance, 'dog' and 'leash' have a strong relationship but are not similar in meaning. Semantic similarity refers to two words sharing semantic properties, for instance 'dogs' and 'cats' which are both animals that people keep as pets (Hill et al., 2015).

### 1.2.2 Semantic priming

In the second experiment, we test whether our VGEs are predictive of semantic priming effects from a large priming experiment (Hutchison et al., 2013). Semantic priming effects occur when activation of a semantically related prime word facilitates the processing of the target word, resulting in shorter reaction times. If all our sensory experiences contribute to word meaning, we would expect visual perceptual properties of the prime-target pair to influence the response times.

Petilli et al. (2021) performed a similar experiment using visual embeddings derived from acti-

vation features from an object recognition network and text-based word embeddings. Their results show that after accounting for the text-based similarity, the visual embedding similarities contribute to explaining the human reaction times only for lexical decision trails with a short stimulus onset asynchrony (SOA), and not for the naming task or long SOA trials. They attribute this to: 1) the lexical decision task being more sensitive to semantic effects than the naming task (Lucas, 2000), and 2) visual information being activated in early linguistic processing and rapidly decaying (Pecher et al., 1984; Schreuder et al., 1998). We will further test these interactions in our own experiment.

## 2 Methods

In our experiments, we compare the VGEs from our own model with three well known text-based distributional semantic models: FastText (Bojanowski et al., 2017), Word2Vec (Mikolov et al., 2013a) and GloVe (Pennington et al., 2014). For the purpose of this study, we take two approaches: 1) we train our own text-based distributional models to allow for a fair comparison to the VGEs, and 2) we use the pretrained models to investigate whether our VGEs capture semantic information that even models trained on large text corpora do not. The code used in this study can be found at https://github.com/DannyMerkx/speech2image/tree/CMCL2022

### 2.1 Training data

MSCOCO is a database intended for training image recognition, segmentation and captioning models (Chen et al., 2015). It has 123,287 images and 605,495 written English captions, that is, five captions paired to each image. Captions were collected by asking annotators to describe what they saw in the picture. Five thousand images (25,000 captions) are reserved as a development set.

The captions are provided in tokenised format. In order to use them in our models we only de-capitalised all words and removed the punctuation at the end of each sentence. This results in a total of 6,184,656 word tokens and 28,415 unique word types, to which we add start- and end-of-sentence tokens for training our visually grounded model.

The images are pre-processed by resizing the images such that the shortest side is 256 pixels, while keeping the original aspect ratio. We take ten 224 by 224 crops of the image: one from each corner, one from the middle and the same five crops for the mirrored image. We use ResNet-152 (He et al., 2016) pretrained on ImageNet to extract visual features from these ten crops and then average the features of the ten crops into a single vector with 2,048 features. These features are extracted by removing ResNet's classification layer and taking the activations of the penultimate layer.

### 2.2 Models

#### 2.2.1 Visually grounded model

Our visually grounded model is based on our implementation presented in Merkx and Frank (2019), and we refer to that paper for the details. Here we will provide a brief overview of the model, any differences with Merkx and Frank (2019) and the parameter settings tested in this study.

The VGE model maps images and their corresponding captions to a common embedding space. It is trained to make the embeddings for matching images and captions as similar as possible, and those for mismatched images and captions dissimilar. The model consists of two parts; an image embedder and a caption embedder. The image embedder is a single-layer linear projection on top of the image features extracted with ResNet-152. We train only the linear projection and do not further fine-tune ResNet.

The caption embedder consists of a word embedding layer followed by a two-layer bi-directional recurrent Long Short Term Memory (LSTM) layer and finally a self-attention layer. The embedding layer has 300 dimensions and is used to represent the input words as learnable embeddings. The purpose of the LSTM is to create a contextualised hidden state for each time-step (input word). Its first layer has 1028 hidden units, while its second layer acts as a bottleneck with 300 hidden units. Finally, the purpose of the attention layer is to weigh each time-step in order to create a single fixed-length embedding for the entire caption. The attention layer has 128 hidden units.

The image embedder has $2 \times 300$ dimensions so that the output matches the size of the caption embeddings. Both image and caption embedding are L2 normalised and we take their distance as the loss signal for the batch hinge loss function (see Merkx and Frank, 2019). The networks are trained for 32 epochs using Adam with a cyclic learning rate schedule based on Smith (2017), which varies the learning rate smoothly between $10^{-3}$ and $10^{-6}$.

The obvious way to extract word embeddings from the trained model would be to use the trained weights of the embedding layer. Unlike for instance in GloVe, where each word's embedding is based on its full co-occurrence distribution, these embeddings are not trained specifically to capture word context or meaning and they are not necessarily the best word embeddings. However, our initial tests showed that they performed very poorly as semantic embeddings when trained from a random initialisation [1]. Rather than taking the input embeddings we create our own embeddings from the hidden representations of the model.

We create our VGEs from the hidden activations of the bottleneck LSTM layer. We use the trained caption encoder to encode all training sentences in MSCOCO. However, we remove the attention layer that creates the sentence embedding and we retain the individual activations of the LSTM at each time step. As the word representations in this layer can be used to create semantic sentence embeddings that capture human intuition about sentence meaning (as we showed for instance in Merkx and Frank, 2019 and Merkx et al., 2021), we expect these representations to better capture word meaning than the input embeddings.

The embedding for each word is then created by summing and normalising its LSTM layer activations from all its occurrences in the dataset. As opposed to Merkx and Frank (2019), where we used a single recurrent layer and found no further benefit of additional layers in terms of sentence embedding quality, we found that the quality of our VGEs improves when we use a two-layer LSTM, with the second layer acting as a bottleneck from which we derive the embeddings.

### 2.2.2 Text-based models

The text-based distributional models are trained on the MSCOCO captions. We train Word2Vec and FastText using the *Gensim* package (Řehůřek and Sojka, 2010). We train GloVe using the code that Pennington et al. (2014) made publicly available[2].

Word2Vec and FastText were trained as the Skip-gram variant with embedding size 300, a context window of 10 and 10 negative samples. GloVe was trained with embedding size 300 and a context window of 10. All resulting word embeddings are

---

[1] Kiela et al. (2018) were able to use the input embeddings because they were initialised using pretrained embeddings.

[2] https://nlp.stanford.edu/projects/glove/

Table 1: Description of the word similarity/relatedness evaluation datasets. #available is the number of word pairs included in the evaluation. Type indicates whether the dataset captures similarity or relatedness. NA indicates subjects were not specifically instructed on the difference.

| Dataset | #word-pairs | #available | type |
|---|---|---|---|
| WordSim353 | 353 | 240 | NA |
| WordSim-S | 203 | 147 | Similarity |
| WordSim-R | 252 | 166 | Relatedness |
| SimLex999 | 999 | 793 | Similarity |
| -SimLex999 Q1 | 249 | 141 | Similarity |
| -SimLex999 Q4 | 250 | 249 | Similarity |
| MEN | 3000 | 2889 | Relatedness |
| RareWords | 2034 | 204 | NA |

then L2 normalised.

In addition, we use the following pretrained vectors (all 300 dimensional): Word2Vec trained on 100 billion tokens of the Google News corpus (Mikolov et al., 2013b), FastText trained on 600 billion tokens of Common Crawl (Mikolov et al., 2018) and GloVe trained on 840 billion tokens of Common Crawl (Pennington et al., 2014).

### 2.3 Evaluation data

#### 2.3.1 Semantic similarity judgements

We include both semantic relatedness and similarity datasets in our analysis. It has been argued that subjects' intuitive understanding of similarity is not necessarily in line with the 'scientific' notions of similarity and relatedness explained in the introduction (Hill et al., 2015). Thus, if subject are not clearly instructed on these notions of similarity or relatedness, we consider the nature of the dataset undefined.

The WordSim353 dataset by Finkelstein et al. (2002) contains 353 word pairs annotated with similarity ratings. While the name suggests it is a similarity rating dataset, more recent studies consider it a hybrid dataset, as subjects were not specifically instructed to judge relatedness or similarity. In a later study by Agirre et al. (2009), the WordSim353 data was split into similar and related pairs by annotating the word pairs. WordSim-S (similar) contains word pairs annotated as being synonyms, antonyms, identical, or hyponym-hyperonym. WordSim-R (related) contains word pairs annotated as being meronym-holonym, and pairs with none of the above relationships but with a similarity score greater than 5 (out of 10). Both sets contain all unrelated words (words not annotated with any of the above relationships and a

similarity lower than 5).

SimLex999 was created with the caveats of the original WordSim353 in mind in order to create a dataset of 999 word pairs annotated for similarity rather than relatedness (Hill et al., 2015). Sim-Lex999 furthermore contains concreteness ratings for the word pairs. Hill et al. (2015) divided the the dataset into concreteness quartiles based on the sum of the concreteness ratings for each pair. Using these quartiles we also look at the 25% most concrete word pairs versus the 25% most abstract pairs in the dataset, of course expecting our grounded model to perform best on the concrete words.

MEN contains 3000 word pairs annotated for semantic relatedness (Bruni et al., 2013). Ratings were collected by showing subjects two word pairs and asking them to select the most related one. MEN was specifically collected to test multi-modal models, by selecting only words that have a visual referent that appeared in a large image database.

The RareWords dataset contains 2034 word pairs, where at least one word of each pair has a low frequency in Wikipedia (Luong et al., 2013). Modelling low-frequency words is a challenge for many models of distributional semantics.

Not all of the words in these databases are available in our training data and thus some will not have a word embedding. Table 1 contains an overview of the datasets described here and the number of word pairs that could be entered in our evaluations.

### 2.3.2 Semantic priming

The Semantic Priming Project (SPP) dataset (Hutchison et al., 2013) contains lexical decision times and naming times from a large priming experiment. The database is large for its kind, with 1,661 target words (and 1,661 non-words for the lexical decision task), each paired with a strong and weak prime and two unrelated primes. Furthermore, each prime-target pair was presented with a short (200ms) and a long (1200ms) SOA. Every combination of prime-target and SOA received responses from 32 subjects.

This gives us 26,576 (1661 target words $\times$ 4 priming conditions $\times$ 2 SOAs $\times$ 2 tasks) trials (disregarding the non-word word trials). We preprocessed the data by removing target words that mistakenly had more or fewer than the required four primes, trials with erroneous responses and missing data. We also lowered any capitals in the prime and target words, averaged the response times over the 32 subjects, and removed any prime-target pair that did not occur in our training data, resulting in 18,326 datapoints.

## 2.4 Analysis

### 2.4.1 Semantic similarity judgements

To test whether the word embedding models capture human intuitions on word similarity, we use the models to calculate embedding cosine similarities for each word pair and correlate them with the human annotations. From the correlations $r$ we derive $R^2$ values, that is, the percentage of variance in the human similarity judgements that is explained by the model similarity scores. This allows us to evaluate our custom trained word embeddings to see which method best extracts word-level semantics from the MSCOCO dataset.

Next, we also compute semi-partial correlations between the human annotations and our VGE model using each of the text-based models as a control. Simply put, the semi-partial correlation between the VGE similarities and human annotations removes the effect of the control (i.e., text-based similarities) from the VGE similarities. Semi-partial $R^2$ gives us the percentage of variance that is uniquely explained by the VGE similarities. Given that all models are trained on the same textual data, with only the VGEs having access to the visual modality, this allows us to see whether visual grounding captures information that the text-based methods do not.

Finally we also test the semi-partial correlations using the pretrained embeddings as a control. For each pretrained model we also add in its custom MSCOCO-trained equivalent as a control, to take into account the information that text-based models can extract from the MSCOCO captions.

### 2.4.2 Semantic priming

Using linear regression models, we analyse how well embedding similarities predict human (log-transformed) reaction times in the SPP data using the Statsmodels package in Python (Seabold and Perktold, 2010). We code SOA and Task as factor variables. The reaction times are not on the same scale due to differences in the required response for the lexical decision and naming tasks so we standardise the log-transformed reaction time data separately for each combination of SOA and Task. This removes the main effects of SOA and Task but we include them in the regression as we are interested in their interactions with the similarity measures.

We fit a baseline regression including the target length (number of characters), Task and SOA as regressors. We furthermore include several regressors based on SUBTLEX-US (Brysbaert and New, 2009): log-transformed word-frequency counts, contextual diversity (the number of SUBTLEX-US documents a word appears in) and the orthographic neighbourhood density (the number of SUBTLEX-US words that are one character edit away) for the target words.

Next, for each of our embedding models, we include the prime-target embedding similarities as a regressor to the baseline model. We also add two two-way interactions to test the claims made in Petilli et al. (2021): 1) the interaction between the embedding similarities and Task to test the difference between lexical decision and naming in terms of sensitivity to semantic effects and 2) the interaction between the embedding similarities and SOA to test their claim about the time-frame in which visual information plays a role. These regression models allow us to compare the word embedding models to each other and to the baseline using the Akaike Information Criterion (AIC), where a lower AIC indicates a better model fit.

We also test if our VGEs can explain variance in the human reaction times that the text-based methods do not. We do this by refitting the regression models for each of the text-based similarity measures and adding the VGE similarity measures and their interactions with Task and SOA as extra regressors. For each of these regressions we then calculate the log-likelihood ratio (LLR) with the corresponding regression without the VGEs, indicating the decrease in model deviance due to adding the VGE similarity measures. Higher LLRs indicate a larger contribution of the VGEs to explaining variance in the human response times beyond what the text-based embedding similarities explain. Because the LLR follows a $\chi^2$ distribution, we can test whether including the VGEs significantly improves the regression model.

We apply a similar approach to the pretrained text-based embeddings, but we also want to account for the information that text-based embedding models can extract from the MSCOCO captions. We do this by fitting a regression model as in the previous step except that we include both the pretrained and MSCOCO trained embeddings and their interactions with SOA and Task. We then follow the same procedure as described above by adding the VGE

similarities and calculate LLRs to see if adding VGEs improves the regression fit.

## 3  Results

### 3.1  Semantic similarity judgements

Figure 1 shows the $R^2$ (explained variance) based on the Pearson correlation coefficients between the human similarity annotations and the embedding similarities. On top of the text-based $R^2$ values, we display the semi-partial $R^2$ of the VGEs using the text-based model as control. As total explained variance equals the semi-partial $R^2$ plus $R^2$ of the control(s), this clearly visualises both the total amount of explained variance and the amount of *extra* variance that is uniquely explained by the VGEs. All Pearson correlations were positive, as expected, except for two non-significant semi-partial correlations which are therefore not included in the figure.

For the MSCOCO models (left panel) we see that while GloVe has the worst performance on each dataset, there is no single best model. Furthermore, while the VGEs are outperformed by FastText and Word2Vec on SimLex999, we see that VGE performs best on the most concrete words (Q4) in SimLex999. A bit surprising then, is that VGE is outperformed by FastText and Word2Vec on MEN, which contains solely picturable nouns.

Looking at the semi-partial $R^2$, that is, the extra variance explained by the VGEs after controlling for one of the other embedding models, we see that for nearly every dataset and every model, the VGEs explain a significant portion of variance that is not explained by the text-based models. This is not very surprising on WordSim, where the VGEs were the best performing embeddings by quite a margin. However, we also see that even though the VGEs are outperformed by FastText and Word2Vec on MEN, they still explain a large extra portion of variance even though the $R^2$ for these models was already quite high.

Lastly, the pretrained models (right panel) outperform the MSCOCO models. This was expected, as the used training data is several orders of magnitude larger than MSCOCO. However, the semi-partial correlations still show that the VGEs explain a significant portion of extra variance on SimLex999 Q4 and MEN.

### 3.2  Semantic priming

The $\Delta$AIC scores in Table 2 show that all word embedding models trained on MSCOCO improve

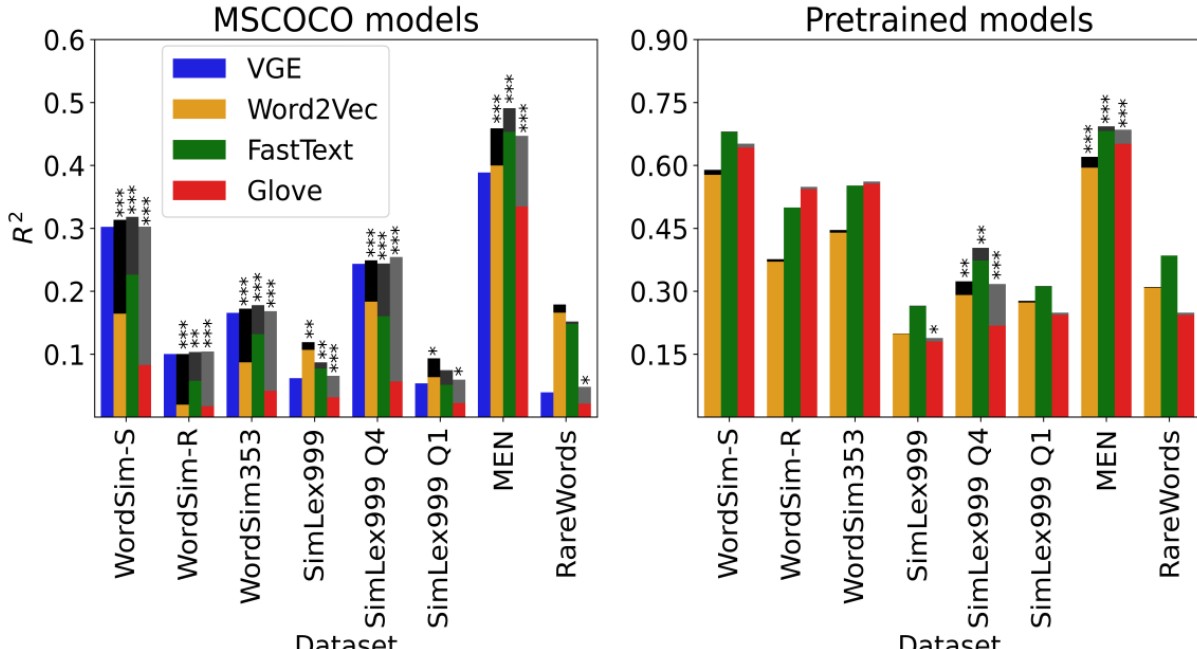

Figure 1: The coloured bars indicate the $R^2$ scores of the four word embedding models. The grey-scale bars on top of the $R^2$ scores of the text-based models indicate the semi-partial $R^2$ scores and their significance ($*p < .05, **p < .01, ***p < .001$, corrected using the Benjamini and Hochberg (1995) procedure with a false discovery rate of 0.05) of the VGEs after controlling for the variance explained by that text-based model. Left panel: models trained on MSCOCO. Right panel: pretrained text-based models.

Table 2: AIC comparison of regression models (lower is better). $\Delta$ indicates the difference in AIC compared to the VGE model or the Baseline model. $\beta$ indicates the coefficient of the embedding similarity main effect (lower is better) and its significance.

| Model | AIC | $\Delta$VGE | $\Delta$Baseline | $\beta$ |
|---|---|---|---|---|
| VGE | 46997.55 | — | $-211.04$ | $-.67^{***}$ |
| FastText | 47101.90 | 104.35 | $-106.86$ | $-.54^{***}$ |
| GloVe | 47163.70 | 166.15 | $-44.88$ | $-.20^{**}$ |
| Word2Vec | 47184.45 | 186.90 | $-24.13$ | $-.22^{**}$ |
| Baseline | 47208.58 | 211.03 | — | — |

Table 3: LLRs between regression models with the indicated text-based similarity measures and the same model with the VGE similarities as extra regressors. $\beta$ VGE are the regression coefficients for the VGE similarities in each model. Higher LLRs indicate a larger improvement in model quality due to adding the VGEs.

| | MSCOCO | | + Pretrained | |
|---|---|---|---|---|
| | LLR | $\beta$ VGE | LLR | $\beta$ VGE |
| Word2Vec | $193.72^{***}$ | $-.77^{***}$ | $69.72^{***}$ | $-.49^{***}$ |
| FastText | $111.46^{***}$ | $-.63^{***}$ | $47.32^{***}$ | $-.42^{***}$ |
| GloVe | $168.34^{***}$ | $-.72^{***}$ | $49.80^{***}$ | $-.36^{***}$ |

the regression fit above the baseline. The embedding similarity effects were all negative, that is, a higher similarity correctly predicts a lower reaction time. We furthermore see that the VGE-derived similarity measures result in the best model fit by quite a margin, as evidenced by the AIC scores and effect size.

We also find significant interactions between Task and the embedding similarities for the VGE ($\beta = 0.201, P = 0.009$) and FastText regression models ($\beta = 0.197, P = 0.027$), meaning that the effect of embedding similarity is stronger for the lexical decision task. We find no significant interactions between the embedding similarities and

SOA.

Table 3 shows the LLRs between regression models including the (pretrained) text-based and our VGE word similarity measures and the corresponding model including only the text-based measures. We see that our VGEs significantly improve the regression fit for every type of text-based method, even when we include both the pretrained and MSCOCO text-based measures. The coefficients of the VGE effects in these models are all positive, meaning a higher VGE similarity predicts a lower reaction time.

In the regression models including the VGEs and the MSCOCO text-based embeddings we found

significant interactions between the VGE similarities and Task in the regression models that also include Word2Vec ($\beta = 0.239, P = 0.007$) or GloVe ($\beta = 0.234, P = 0.01$) and no other interactions with Task or SOA.

Lastly, in the regression models including the VGEs and both pretrained and MSCOCO text-based embeddings, we find significant interactions with Task for Word2Vec ($\beta = 0.312, P < 0.001$), FastText ($\beta = 0.297, P = 0.001$) and GloVe ($\beta = 0.443, P < 0.001$) vectors, and none for the VGEs.

## 4  Discussion

We created Visually Grounded Embeddings using a caption-image retrieval model in order to test if these embeddings can capture information about word meaning that text-based approaches cannot. Importantly, by testing our VGEs on human behavioural measures typically thought to rely on conceptual/semantic knowledge, we test a central idea of embodied cognition theory, namely that our visual experiences contribute to our conceptual knowledge.

### 4.1  Semantic similarity judgements

Our first experiment showed that, when trained on the same corpus, our VGEs are on par with text-based methods. While there is no clear overall best method, the VGEs perform well on WordSim and, as might be expected, on the datasets with concrete picturable nouns. Even though the text-based methods outperform the VGEs on one of these (MEN), the VGEs still explain a significant amount of extra variance over and above what is explained by the text-based methods. This indicates that the text-based embeddings and VGEs capture non-overlapping conceptual knowledge, which we attribute to the visual grounding of the VGEs, given that the training materials were otherwise equal.

The only database where the VGEs performed notably worse than the text-based methods was RareWords. This is perhaps because during training, the VGEs are grounded in the image corresponding to the text input, even if not all words in the sentence are visible in the picture. As the words in RareWords are generally not picturable nouns, any visual information incorporated into the word-embedding is unlikely to be helpful, or, as evidenced by the results, counterproductive.

We furthermore found that our VGEs explain additional variance in the human similarity ratings even after accounting for both the MSCOCO text-based models and pretrained models trained on massive text corpora. The fact that the VGEs explain a significant amount of extra variance even after the text-based models have seen billions of tokens of text, suggests that some aspects of word meaning cannot be captured solely from text and as well as that visual similarity plays a role in human intuition about word meaning.

### 4.2  Semantic priming

In our second experiment, the VGEs outperformed the text-based methods on explaining human reaction times from the Semantic Priming Project. Even after we account for both the MSCOCO text-based models and pretrained models in our regression, the VGEs still explain a significant amount of variance in the reaction times.

In previous work, Petilli et al. (2021) only found a significant contribution of visual information in the short SOA lexical decision task. We found no further proof for their hypothesis that visual information is activated in early linguistic processing and thereafter rapidly decays. Rather, we find that our VGEs improve the model quality for both short and long SOA trials.

We did find a significant positive interaction with Task, meaning that the word embeddings explain less variance in the naming task than in the lexical decision task. This interaction was not specific to the VGEs but also occurred in the models including FastText and for all the pretrained embeddings. As claimed in Petilli et al. (2021) and Lucas (2000) this suggests that naming tasks are in general less sensitive to semantic effects.

## 5  Conclusion

We set out to test an end-to-end approach to combining visual and textual input in a single embedding, trained on a cognitively plausible amount of data. The results from our two experiments suggest that VGEs capture aspects of word meaning that text-based approaches cannot. Even though we include word embeddings trained on corpora several orders of magnitude greater than any human's exposure to language, our VGEs still explain a unique portion of variance in both human behavioural measures.

While our results indicate that visual grounding can provide complementary information for certain

words, it may not play a role in our conceptual knowledge of rare, abstract words, as shown by our results on the RareWords corpus. Similar to Petilli et al. (2021) this then does not support the strongest formulations of embodied cognition theory which suggest total equivalence between conceptual and sensorimotor processing (Glenberg, 2015).

Of course, one could always claim that it is just current word-embedding models that do not fully capture word meaning yet. However, given that VGEs trained on a relatively small amount of visual data can complement text-based embeddings, we do not think even larger text-corpora or more complex embedding models can ever fully capture human semantic knowledge. The human experience is rich and varied, and our computational models can never fully capture human word knowledge while ignoring visual aspects of this experience.

## 6 Acknowledgements

The research presented here was funded by the Netherlands Organisation for Scientific Research (NWO) Gravitation Grant 024.001.006 to the Language in Interaction Consortium.

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
