# OpenReview forum: "Seeing the advantage: visually grounding word embeddings to better capture human semantic knowledge"
_aclweb.org/ACL/2022/Workshop/CMCL — CMCL 2022_

### Official Review · Reviewer_P9g8 · 2022-03-21
**The paper presents visually grounded word embeddings and show that they predict both human reaction times and human judgements of word similarity. This is an interesting work that very well fits the theme of the workshop.**

**Rating:** 8
**Confidence:** 4

**Review:**

The paper presents visually grounded word embeddings and shows that they predict both human reaction times and human judgements of word similarity. I like the fact that the paper compares both to a reaction time experiment  and to word similarity judgement data. The fact that the proposed model performs well no both, is a nice indication of its cognitive plausibility.

---

### Official Review · Reviewer_wj8M · 2022-03-25
**Careful experiments that disentangle visual information in text and actual image data**

**Rating:** 7
**Confidence:** 4

**Review:**

This is a well-written paper that carefully experiments with VGEs and compares the additional variance that can be explained using VGEs compared to text-based models.


Strong points:

The paper makes a contribution to previous work by disentangling the visual information that may be contained in the text part of a dataset such as MSCOCO and the actual image data.


It shows results on a large number of datasets and tasks and meaningful subparts thereof. There is definitely substance.


Weak points:

The paper investigates a worthwhile but considerably researched topic on two  well-known tasks. The method for creating VGEs is taken from previous work with some adaptations. In terms of novelty, the paper does not excel.


Even after careful re-reading, I am not completely convinced that the results between the VGE model and the textual models are comparable. The method used to create the VGE is so different (more contextualised than the text-based models) that one wonders whether the contributions of the VGE model are not due to the model instead of the visual features.


I would also like to see Figure 1 better explained, in particular the grey bars. These represent ‘the partial R2 of the VGEs after controlling for the variance explained by that text-based model’. Do you mean here the given text-based model, so word2vec for example? From the way these scores are represented, one on top of the other, you would get the idea that these scores can be added to arrive at a combined performance. I doubt that this is true though. Perhaps the partial R2 scores could be explained in more detail.

---

### Official Review · Reviewer_4sDq · 2022-03-27
**Combining vision and language to model semantic similarity and relatedness**

**Rating:** 6
**Confidence:** 4

**Review:**

In this paper, the authors investigate the use of multimodal (text+image) embeddings to predict semantic similarity and relatedness between word pairs.
All in all, the paper is well written and discusses a topic that is central to the CMCL community. However, there are still parts that are unclear and this makes the overall contribution weaker.

* It is not clear to me how you can say that: VGEs explain additional variance after including text-based representations? This really needs more explanations.

* Similarly, it is not clear how you can quantify the non-overlapping information captured by the different models.

* Figure 1 needs more attention. Firstly, it has to be explained in more detail. Moreover, the two scales should be the same: this is extremely misleading.

---

### Decision · Program_Chairs · 2022-03-29

Accept